# Dental microwear texture analysis reveals behavioural, ecological and habitat signals in Late Jurassic sauropod dinosaur faunas

**Daniela E. Winkler** [1,2,8] ✉, **Emanuel Tschopp** [2,3,4,5,8] ✉, **André Saleiro** [6,7], **Ria Wiesinger**[3] & **Thomas M. Kaiser**[2,3]

Most faunas from the Mesozoic era were dominated by sauropod dinosaurs, the largest terrestrial animals to ever exist. These megaherbivores were remarkably diverse and widely distributed. Here we study three Late Jurassic faunas from the USA, Portugal and Tanzania, each approximately 150 million years old, which are known for their extreme sauropod diversity. Whereas general taxonomic composition was similar in these three faunas, the major clades differed in relative abundance. Moreover, their depositional strata record distinct climatic regimes. Using dental microwear texture analysis, we investigated the impact of these climate regimes and the resulting food availability on the different sauropod taxa. Wear patterns in camarasaurid macronarians show minimal variation across different climate regimes, supporting previous studies suggesting that these animals migrated to follow their preferred climate niche and food source. North American camarasaurids show similar wear patterns to those of Portuguese turiasaurs, another broad-crowned taxon, which did not exist in the Jurassic of North America. By contrast, where camarasaurids and turiasaurs co-occurred in Portugal, their microwear patterns are distinct, suggesting niche differentiation to avoid ecological competition between these two clades. Flagellicaudatan diplodocoids display highly variable wear patterns, indicating limited migration (and therefore seasonal variation in diet), which aligns with observed biogeography patterns in the USA. Early-branching titanosauriforms show highly distinct wear patterns between different climate regimes, which can probably be attributed to different abrasive loads in the respective habitats. Our results demonstrate that dental microwear texture analysis not only records dietary preferences but also reveals behaviour such as competition and migration related to dietary niches in past ecosystems.

[1]Zoological Institute, Kiel University, Kiel, Germany. [2]Centre for Taxonomy and Morphology, Leibniz Institute for the Analysis of Biodiversity Change, Museum of Nature, Hamburg, Germany. [3]Fachbereich Biologie, Universität Hamburg, Hamburg, Germany. [4]Freie Universität Berlin, Institut für Geologische Wissenschaften, Berlin, Germany. [5]Present address: Department of Vertebrate Paleontology, American Museum of Natural History, New York, USA, NY. [6]GeoBioTec, NOVA School of Science and Technology, University NOVA of Lisbon, Caparica, Portugal. [7]Museu da Lourinhã, Lourinhã, Portugal. [8]These authors contributed equally: Daniela E. Winkler, Emanuel Tschopp. ✉e-mail: dwinkler@zoologie.uni-kiel.de; e.tschopp@fu-berlin.de

Late Jurassic ecosystems across the planet were dominated by sauropod dinosaurs in terms of size and body mass contribution[1]. The three best-known and represented Late Jurassic faunas are from the USA, Tanzania and Portugal. These sauropod faunas had very similar taxonomic compositions, with brachiosaurid macronarians and diplodocoids occurring in all three. The North American and Portuguese faunas furthermore shared the presence of camarasaurid macronarians, whereas turiasaurs and early somphospondylans occur both in Portugal and Tanzania. The Tanzanian fauna further included non-neosauropod mamenchisaurids[2–7], which are otherwise primarily known from the Jurassic of Asia[8–11]. Although generally similar, within-clade species diversity and the abundance of individuals from the distinct taxa are different between the three faunas. Whereas camarasaurid macronarians and diplodocid diplodocoids dominated in the USA[12–15], brachiosaurid titanosauriforms and dicraeosaurid diplodocoids predominated in Tanzania[5,16] and turiasaurs in Portugal[17,18]. The reasons for these unequal distributions of taxon abundance remain poorly understood.

As the largest terrestrial megaherbivores that ever roamed the Earth, sauropods must have greatly depended on plant productivity and availability. Similarly to sauropods, major plant groups were distributed globally during the Late Jurassic, with conifers forming the canopy together with ginkgoes. Tree ferns reached intermediate heights, whereas other ferns, seed ferns, cycads and horsetails usually composed the understorey. The only notable difference between the floras of the three sauropod-bearing formations was the absence of the conifer clade Pinaceae and of horsetails in Tendaguru, where also cycads were a comparatively minor component of the plant cover[19–21]. Moreover, the relative abundance of the different plant clades differed between the three biomes[21], which was correlated to distinct climate regimes.

Palaeoclimatic reconstructions indicate different climatic conditions in these three regions during the Late Jurassic[20,22–25]. The western USA was reconstructed as semi-arid to arid, with a mean annual temperature ranging between 12 °C and 30 °C, depending on latitude[22,25] and probably strong seasonality[24]. Mean annual precipitation was found to be higher in Portugal and Tanzania compared with the USA, suggesting a more humid climate than in the USA[24,25]. In Portugal, mean annual temperature ranged between 18 °C and 24 °C, whereas it was considerably warmer, with 24–30 °C, in Tanzania[25]. Seasonality was probably strong in Tanzania as well, with wet winters[22,26] and dry summers[20], corresponding to a monsoon-type climate[25]. We hypothesize that these different climate regimes probably controlled forage availability, which in turn shaped distribution and abundance of different sauropod taxa adapted for foraging on specific plant taxa. Furthermore, we propose that large-bodied herbivores in such habitats affected by seasonality either adopted a generalist feeding strategy, using diverse dietary resources in the same place year-round, or depended upon seasonal migration if they had a narrower dietary niche.

Strong niche partitioning between major sauropod groups has been suggested on the basis of their distinct skull shapes and tooth morphologies, and the posture of the neck and forelimbs[7,18,27–40]. However, morphology can provide only general information about possible feeding adaptations and is not direct evidence whether a certain dietary resource was actually exploited. A promising avenue to assess niche partitioning in sauropods and other archosaurs is the study of dental microwear as a dietary proxy[35,41–46]. We here use three-dimensional dental microwear texture analysis (DMTA), a semi-automated quantitative approach to evaluate microscopic surface wear of enamel wear facets[47,48], as a means to test whether distinct sauropod taxa occupied the same niche in three different geographical areas (Fig. 1) irrespective of climate, and how these food preferences and availability may have shaped sauropod distribution during the Late Jurassic period.

## Results

We analysed measurements taken from the enamel of buccal and occlusal surfaces from the sauropod dental microwear texture (DMT) dataset acquired by ref. 49, which were categorized as either scan quality 1 or 2 (see Methods and refs. 49,50 for details). Because ref. 50 found that measurements from casted tooth surfaces in this dataset yielded inaccurate results, we only exported measurements taken from original teeth and from moulds. Furthermore, as we aimed to compare the impact of diverging climatic regimes on the well-studied faunas of the Morrison (USA), Tendaguru (Tanzania) and Lourinhã (Portugal) Formations, we excluded all measurements of teeth from other geographical areas. The final set of 322 measurements belongs to 39 sauropod individuals, 17 of which were recovered from the Lourinhã Formation (Portugal; all represented by isolated teeth), 13 from the Morrison Formation (USA; between 1 and 8 teeth per specimen) and 9 from the Tendaguru Formation (Tanzania; between 1 and 8 teeth per specimen) (Supplementary Table 1). They include teeth from a range of tooth positions and wear stages, but these differences are expected to have minimal impact on the final results because (1) there are relatively few morphological differences along the tooth rows of sauropods[7,18,32,51,52] and (2) based on studies on mammalian dental microwear, recorded wear features in the tooth enamel reflect the ingesta of the past few weeks to months irrespective of how worn they already are[53–55].

Despite our sampling efforts, the three geological formations and four clades could not be equally represented. Owing to the difference in sample size and distribution of clades, our study sample allows for limited direct comparisons of clades in different faunas and climate regimes (Fig. 1). Turiasaur samples were restricted to Portugal. Camarasaurids and flagellicaudatans were sampled from Portugal and the USA. Titanosauriforms could be compared between Portugal and Tanzania. For representative images of typical teeth and analysed microwear patterns for each taxon and geological formation, see Extended Data Fig. 1.

First, we tested whether buccal and occlusal surfaces could be compared directly. Most height (matf, metf, Sa, Sdc, Sk, Sp, Sq, Sv, Sz; see Supplementary Table 6 for parameter descriptions) and volume (Vm, Vmc, Vv, Vvc, Vvv) parameter values were larger for occlusal surfaces than for buccal surfaces in Titanosauriformes, Camarasauridae, Turiasauria and an unidentified macronarian (Fig. 2 and Supplementary Table 1). Complexity (Asfc, Sdr) and mean slope (Sdq) parameters showed the same pattern. Mean density of furrows (medf) was comparable for occlusal and buccal surfaces in Camarasauridae and Turiasauria, but it was lower for occlusal surfaces than buccal surfaces in Titanosauriformes. For Flagellicaudata, only four scans of one occlusal surface were of good enough quality for the analyses, all other surfaces were buccal. Height and volume parameters were very similar between buccal and occlusal surfaces, whereas complexity (Asfc, Sdr) and density of furrows (medf) were higher for buccal surfaces than for occlusal surfaces in Flagellicaudata (opposite to macronarians and turiasaurs). Given the ambiguous results, we performed the subsequent analyses both with separate datasets for occlusal and buccal surfaces, as well as with the combined dataset.

Of all major clades, flagellicaudatans show the highest variability across all geographical regions and in nearly all parameter values (Supplementary Fig. 1). Sampled titanosauriforms from Tanzania (which are probably all brachiosaurids; E.T., personal observation, 2022) are highly variable as well, whereas camarasaurids and turiasaurs occupy very restricted areas in the principal component analysis (PCA) plots (Fig. 3). In a canonical variate analysis (CVA), taxa could mostly be separated with little overlap and confident assignment to the correct clade (Fig. 4b and Supplementary Table 2). However, Portuguese titanosauriforms and turiasaurs strongly overlap with North American camarasaurids, and to a lesser extent with flagellicaudatans from Portugal (Fig. 4a). These four groups are all distinct from Portuguese camarasaurids, North American flagellicaudatans and brachiosaurids from Tanzania (Fig. 4).

Comparing taxa measured from two different geographical regions, flagellicaudatans from Portugal had distinctly lower values related to wear surface complexity, slope and density of furrows, than North American flagellicaudatans. Titanosauriforms from Tanzania

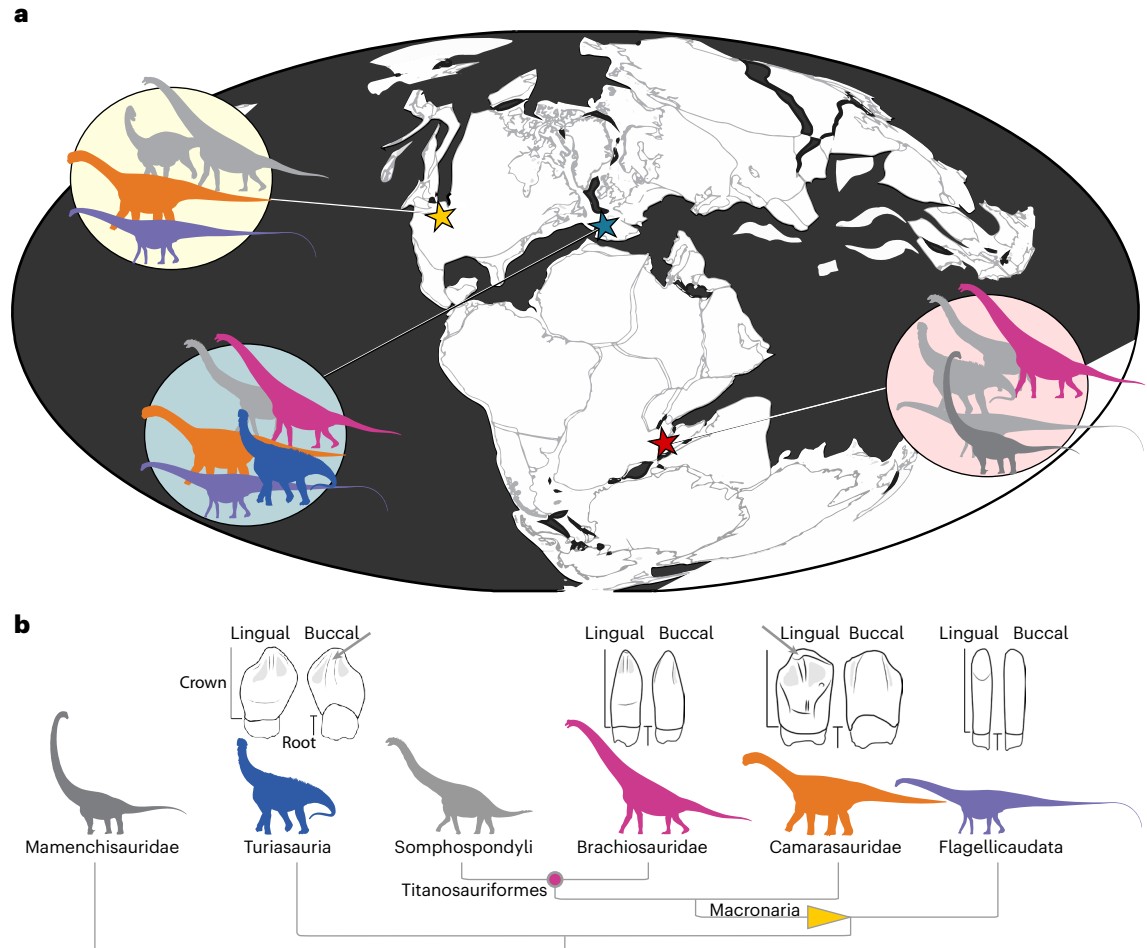

**Fig. 1 | Sauropod diversity and tooth morphology at the sampled locations.**
**a**, Palaeogeographic map showing the approximate arrangements of continents during the Late Jurassic. Yellow star, North America (Morrison Formation); blue star, Portugal (Lourinhã Formation); and red star, Tanzania (Tendaguru Formation). Silhouettes represent major taxonomic groups of sauropods present in the sampled locations. Taxa that were not available for inclusion into the current study from the respective locations are shown in light grey, those included in the analysis are shown in colour. **b**, Tooth morphologies for selected clades and phylogenetic tree. All clades except possibly Somphospondyli and Mamenchisauridae could be sampled for the current study. The indeterminate sauropods and macronarians sampled in this study could not be attributed to a less-inclusive clade. Tooth morphologies are shown for well-represented clades (from left to right, Turiasauria, Titanosauriformes, Camarasauridae, Flagellicaudata). Teeth are shown in lingual and buccal view. Approximate sampling areas for occlusal and buccal surfaces are shown in camarasaurid and

turiasaur teeth, respectively (grey arrows). Credits: **a**, Maps are taken from The Paleobiology Database Navigator (https://paleobiodb.org/navigator/) under a CC BY 4.0 license, which uses GPlates as a data source for the maps. GPlates are shared under the GNU software general public license, v.2 (https://www.gnu.org/licenses/old-licenses/gpl-2.0.html). **a,b**, Silhouettes are from Phylopic (https://phylopic.org). *Xinjiangtitan shanshanesis* (Mamenchisauridae), created by Jagged Fang Designs under a CC0 1.0 license; *Haplocanthosaurus priscus*, created by T. M. Keesey under a CC0 1.0 license; *Amanzia greppini* (Turiasauria), created by T. Dixon under a CC BY 4.0 license; *Euhelopus zdanskyi* (Somphospondyli), created by DiBgd and modified by T. M. Keesey under CC BY-SA 3.0 license; *Giraffatitan brancai* (Brachiosauridae), created by S. Hartman under a CC BY 3.0 license; *Diplodocus carnegii* (Flagellicaudata), created by S. Hartman under a CC BY 3.0 license; *Camarasaurus supremus* (Camarasauridae), created by M. Wedel under a CC BY 3.0 license. Tooth shapes are modified from ref. 18, Wiley.

tend to show larger height and volume parameters, as well as higher complexity values, and significantly larger slope and density than those from Portugal (Fig. 5 and Supplementary Table 4). Camarasaurids from Portugal and the USA are similar in parameter values, except for height and volume parameter values, which are higher in Portuguese camarasaurids compared with those from the USA (Fig. 2 and Supplementary Table 4). Different regions (and thus climate regimes) also seem to have had a distinct impact on wear patterns of their megaherbivore inhabitants. When comparing only buccal surfaces, visible, non-significant differences exist between flagellicaudatans from the USA and Portugal and between titanosauriforms from Tanzania and Portugal. Camarasaurids from the USA and Portugal are very similar in most DMT parameters (Fig. 5, Supplementary Fig. 1 and Supplementary Table 4). When compared with brachiosaurids from Tanzania, significant differences are found with camarasaurids from

both Portugal and the USA (Fig. 5) as well as with flagellicaudatans from Portugal (Supplementary Table 4).

In Portugal, no differences are seen among taxa regarding complexity parameters (Asfc, Sdr). These parameters, however, are clearly distinct between flagellicaudatans (larger and more variable) and camarasaurids (lower and more consistent) in the USA. Slope (Sdq) and density of furrows (medf) followed a similar pattern to complexity parameters, with little differences between all sauropod clades from Portugal, whereas in the USA, flagellicaudatans showed distinctly larger values than camarasaurids.

Only limited comparisons can be made within the Tanzanian sample, because the vast majority of sampled teeth were from brachiosaurid titanosauriforms. Only an indeterminate sauropod tooth and an indeterminate macronarian survived the dataset cleansing for the final analyses. Whereas the absence of enamel wrinkling on

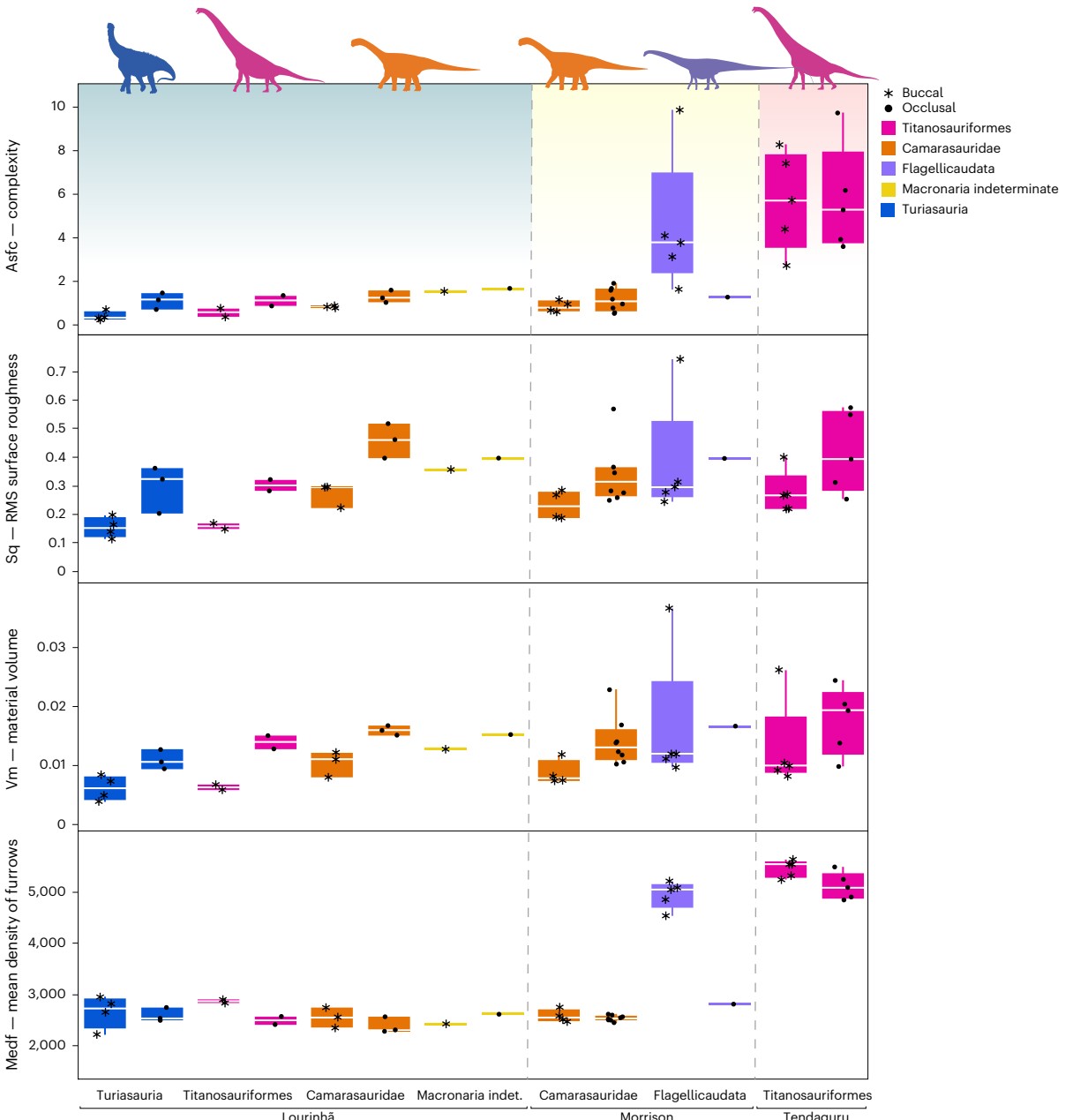

**Fig. 2 | Selected DMT parameters representing complexity, height, volume and density for buccal and occlusal surfaces of the clades where both surfaces could be measured.** Owing to the distinct differences between buccal and occlusal surfaces for all parameters and in most clades, we separate surfaces in analyses concerning diet reconstruction. Note that Flagellicaudata from Portugal, unidentified Macronaria from the USA, and unidentified Sauropoda and Macronaria from Tanzania are missing as they are only represented by either buccal or occlusal surfaces. RMS, root-mean square. Boxplots are depicted with a thick horizontal bar that represents the median; the box encloses the first (25%) and third (75%) quartiles; the whiskers extend to the full interquartile range. Turiasauria, $n = 5$; Titanosauriformes, $n = 9$; Camarasauridae, $n = 12$; Flagellicaudata, $n = 6$; and Macronaria indet., $n = 1$. Credit: Silhouettes are from Phylopic (https://phylopic.org). *Amanzia greppini* (Turiasauria), created by T. Dixon under a CC BY 4.0 license; *Giraffatitan brancai* (Titanosauriformes), created by S. Hartman under a CC BY 3.0 license; *Diplodocus carnegii* (Flagellicaudata), created by S. Hartman under a CC BY 3.0 license; *Camarasaurus supremus* (Camarasauridae), created by M. Wedel under a CC BY 3.0 license.

the indeterminate sauropod tooth suggests a basal, non-eusauropod affinity (which is surprising because such early forms are currently not represented by other skeletal material in the Late Jurassic biomes discussed above)[5,7,14,18,56], the indeterminate macronarian tooth could stem from a brachiosaurid titanosauriform. The single occlusal surface from the indeterminate sauropod and the single buccal surface of the macronarian fell within the range of brachiosaurid buccal surfaces, so all specimens from Tanzania showed similar values within the fauna.

Overall, there seems to be a strong location-specific tendency consistent across taxa (Fig. 6 and Supplementary Table 5). Pooled specimens from Tanzania are characterized by significantly larger complexity (Asfc, Sdr) and density of furrows (medf) when compared with pooled specimens from both Portugal and the USA. This location-specific separation also becomes evident in the PCAs, which show a clear separation of the Tanzanian fauna from the Portuguese fauna. The assemblage from the USA partially overlaps with Tanzania when buccal or buccal and occlusal surfaces are considered but is completely separated when only occlusal surfaces are considered. USA and Portugal greatly overlap, regardless of whether occlusal, buccal or all surfaces are compared.

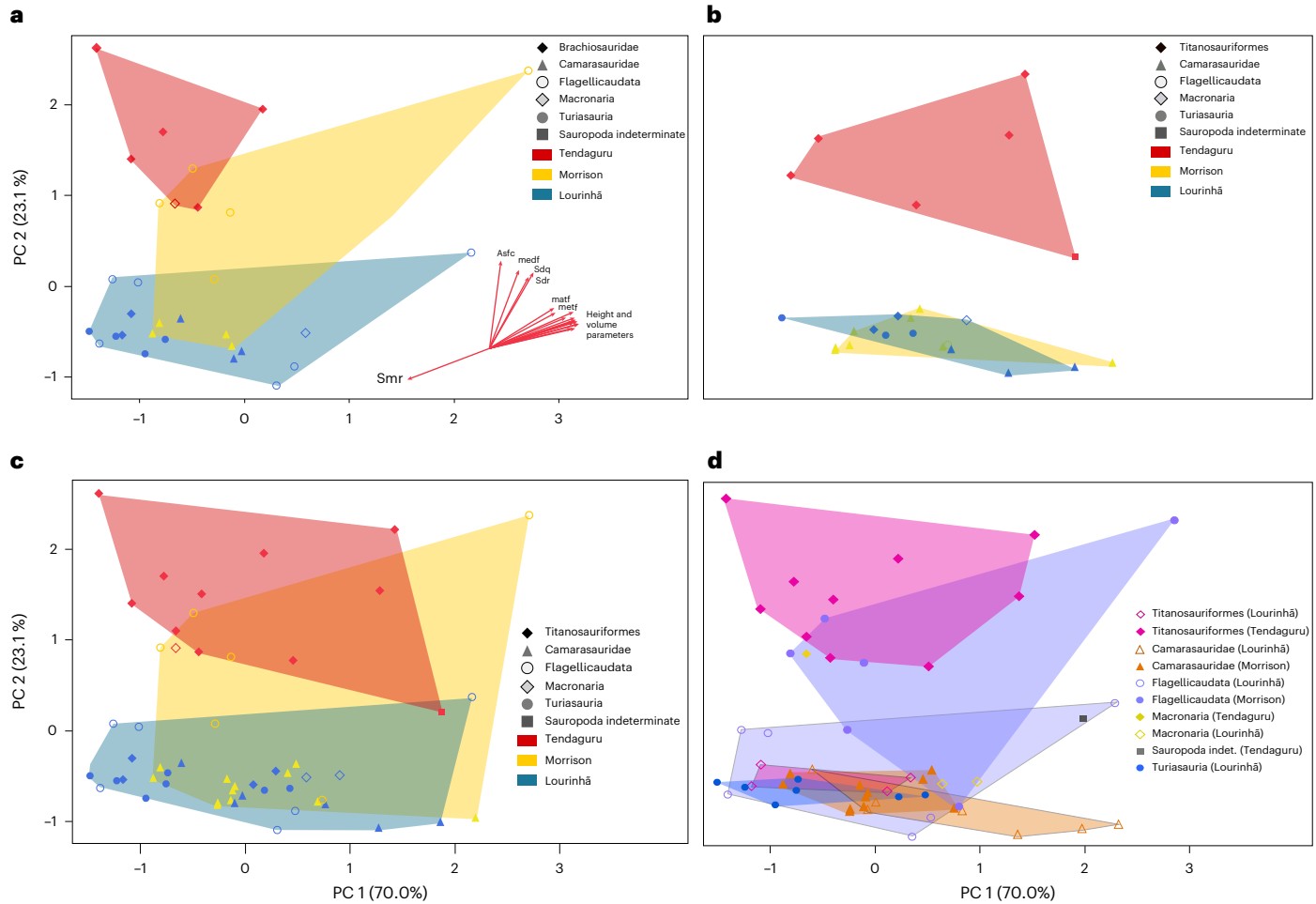

**Fig. 3 | PCA using 20 DMT parameters. a**, Buccal surfaces only. **b**, Occlusal surfaces only. **c**, Buccal and occlusal surfaces combined. **d**, As in **c** but separated according to clade.

## Discussion

DMTA revealed dietary overlap between some sauropod clades, but also distinct niche segregation between others. Moreover, ecosystem (fauna)-specific wear signatures indicate that habitat parameters in the three analysed Late Jurassic ecosystems from Portugal (Lourinhã Formation), USA (Morrison Formation) and Tanzania (Tendaguru Formation) were different, probably owing to climatic differences, which affected both plant community and environmental abrasive load.

### Dental morphology and oral food processing behaviour affect DMT

Overall higher roughness and complexity were observed in occlusal versus buccal surfaces in all taxa except for Flagellicaudata, where occlusal surfaces could only be observed in one tooth specimen. Hence, the limited occurrence of flagellicaudatan occlusal surfaces in our dataset hampers direct comparison with taxa that had tooth-to-tooth occlusion (and mammaliaform comparisons). The differences in occlusal versus buccal wear patterns are probably related to the different tooth morphologies. Turiasaurs and early-branching macronarians (represented by camarasaurids, brachiosaurids and possibly early somphospondylans in our dataset; Fig. 1), show true occlusion between antagonistic teeth. Such tooth-to-tooth occlusion resulted in strongly developed, V-shaped wear facets reaching from the apex down the mesial and distal surfaces[7,17,18,27,28,35,51]. Although the teeth lack adaptations for advanced oral food processing, such as chewing, such attritional contacts increase general wear on the occlusal surfaces compared with buccal surfaces. Through attritional contacts and tooth-to-tooth interaction,

food is also trapped between teeth more efficiently, resulting in more tooth-to-food contacts (abrasion). Buccal surfaces hence experience less food contacts than occlusal surfaces, which is also evidenced by the development of the distinct occlusal wear facets with exposed dentin. In flagellicaudatans, the very narrow, pencil-shaped teeth show no distinct occlusal contacts, and only a single wear facet with exposed dentin formed on the tip of the tooth, which can be considered 'occlusal surface-like' (compare Fig. 1).

### Ecological exclusion between taxa with pronounced tooth-to-tooth occlusion

Camarasaurids and turiasaurs had very similarly shaped, broad-crowned teeth with well-developed tooth-to-tooth occlusion[7,17,18,51] and relatively narrow snouts[57,58]. Although these overlapping morphologies may suggest similarities in feeding niche, there was a distinct separation between camarasaurids and turiasaurs co-occurring in Portugal based on DMT data (Fig. 4a). North American camarasaurids, on the other hand, show considerable overlap with Portuguese turiasaurs. They show consistently low (height and roughness) signals, which is interpreted as a low-abrasive, consistent diet. In generalists, we would expect greater diversity of incorporated forage plants, and hence also a less consistent, more variable DMT signal. Portuguese camarasaurids are slightly different from their counterparts from the Morrison Formation, showing overall larger surface roughness. This pattern is best explained by ecological competition between camarasaurids and turiasaurs where they co-occurred in Portugal, which resulted in niche partitioning. Our data suggest that camarasaurids in Lourinhã

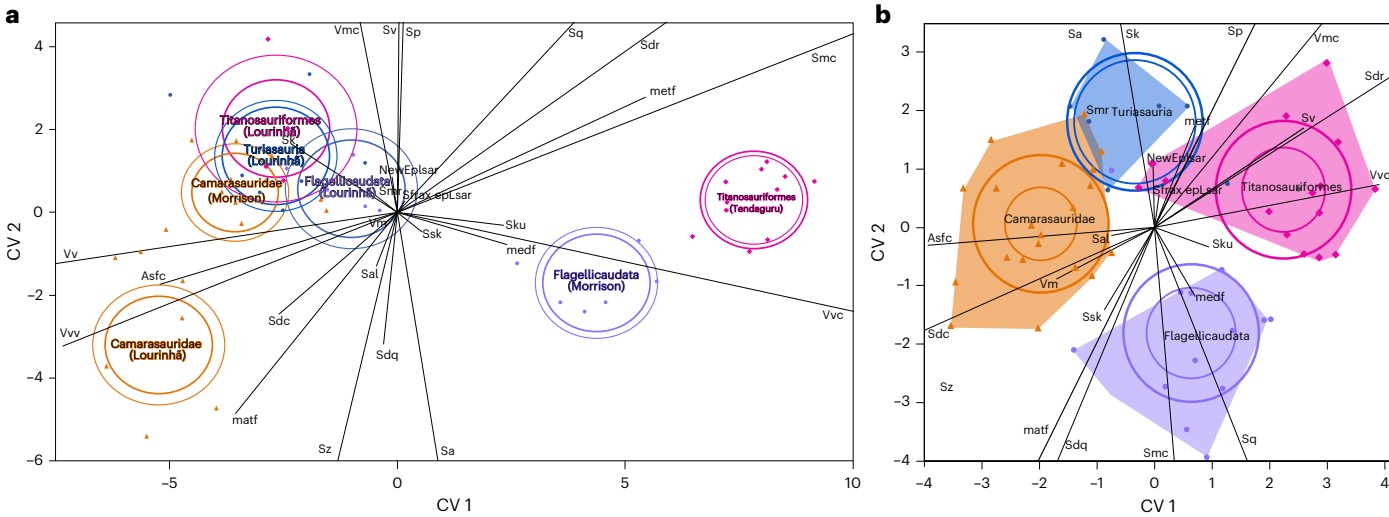

**Fig. 4 | CVA including all DMT parameters. a**, Assignment according to clade and fauna. **b**, Assignment according to clade only. Note that Camarasauridae from the Morrison Formation are greatly overlapping with Turiasauria from Lourinhã and with Titanosauriformes from Lourinhã. Unidentified Macronarians and Sauropoda indet. are excluded.

ended up occupying a different dietary niche from turiasaurs, and from contemporary camarasaurids, in North America. The absence of turiasaurs in the North American Morrison Formation allowed camarasaurids on that continent to occupy a similar feeding niche as that occupied by turiasaurs in Portugal (Fig. 4a). The unique dietary niche of camarasaurids in Lourinhã indicates that camarasaurids were more flexible in their dietary traits than turiasaurs, or that turiasaurs outcompeted camarasaurids in their preferred niche in Portugal—or a combination of both. At the same time, camarasaurids from North America and Portugal were similar in their very low variability of wear features.

**Variability in DMT patterns hints at migration behaviour**

Additional evidence for migratory behaviour in sauropods derived from dental wear studies is important, as the currently available evidence is so sparse. On the basis of stable isotope data from a single *Camarasaurus* tooth from the Morrison Formation, ref. 59 suggested that this animal migrated throughout the year, potentially following its preferred food source and/or climate niche over the seasons. In fact, the strong seasonality in the Morrison Formation habitat[24] probably impacted availability of a herbivore's preferred forage. A seasonal signal is known from microwear in mammals[53–55], and must have been present in sauropods, too, given their high tooth formation and replacement rates[52,60–62], which led to very short exposure times of a few weeks to maximum 2 or 3 months for each functional tooth. Therefore, if seasonal dietary shifts existed, finding a consistent dental microwear signal within one sauropod taxon is unlikely, as it would imply that all sampled teeth were derived from individuals that died in the same season. The consistently low variation in the DMT signal of camarasaurids from both the Morrison and the Lourinhã Formations is hence remarkable. It hints at a very narrow dietary niche occupied by camarasaurids, which may have forced these taxa to seasonally migrate and follow climate-driven availability patterns of their preferred food source throughout the seasons—not only in northern America but also in Portugal. Their absence from the Tendaguru fauna[5] could indicate that camarasaurids were not well-adapted for the hot, tropical, and presumably less seasonal, climate of the region.

The strongly variable signal in flagellicaudatans of the Morrison and Lourinhã Formations, in combination with even faster tooth replacement rates than camarasaurids[52,60], on the other hand, hints at a broad dietary niche and rather unselective foraging. Unselective foraging has also been inferred for at least adult flagellicaudatans on the basis of their squared snout shape[35]. Such a broad dietary niche, indicated by variable DMT and the presence of differently sized wear features, would have allowed these animals to shift their diet to adapt to seasonal availability of resources. A seasonal dietary shift would render migration in this clade less necessary. Non-migratory behaviour in flagellicaudatans is also in parts supported by the biogeographic pattern of these clades within the Morrison Formation, where macronarian genera generally seem to have been more widespread compared with flagellicaudatan genera[13,15,63,64]. The more global distribution of macronarians within the formation may then have been partly due to their migratory behaviour, whereas flagellicaudatans restricted their geographic range by not migrating. The different environments they inhabited seem to have impacted microwear patterns, too.

**Abrasive loads of different environments**

The strongly overlapping ranges of Portuguese titanosauriforms and turiasaurs in the CVA (Fig. 4), controlled by their high similarity in general surface roughness and complexity (Supplementary Fig. 1), show that these clades cannot securely be distinguished on the basis of DMT patterns. Hence, there seems to be a very strong environmental signal that overprints any dietary niche signal in Portugal, at least in these two clades, no strict niche separation between them or a distinct niche separation that could not be captured by DMTA or within the available dental sample. The overall lower values for height, volume and complexity parameters as well as low density of furrows for taxa from Portugal as compared with the Tanzanian fauna indicate that a low-abrasive diet was abundantly available in Portugal. Portuguese titanosauriforms, camarasaurids and turiasaurs are all very similar in their DMT patterns (Fig. 2 and Supplementary Fig. 1). If these taxa were all migrating to some extent, following their preferred low-abrasive food sources, they must have occupied large enough home ranges to avoid competition—which seems to be supported by their large body size[65]. Camarasaurids from the USA showed very similar dental wear to turiasaurs (and also camarasaurids, to some degree) from Portugal (Fig. 4a). The Tanzanian taxa, however, are significantly different (compare Fig. 6). Brachiosaurid titanosauriforms dominate our sample from Tanzania, and the combined sauropod fauna from Tendaguru show significantly larger complexity and density of furrows as compared with the faunas from both Portugal and the USA. Therefore, the sauropod taxa from the Tendaguru ecosystem must have differed in feeding preferences and/

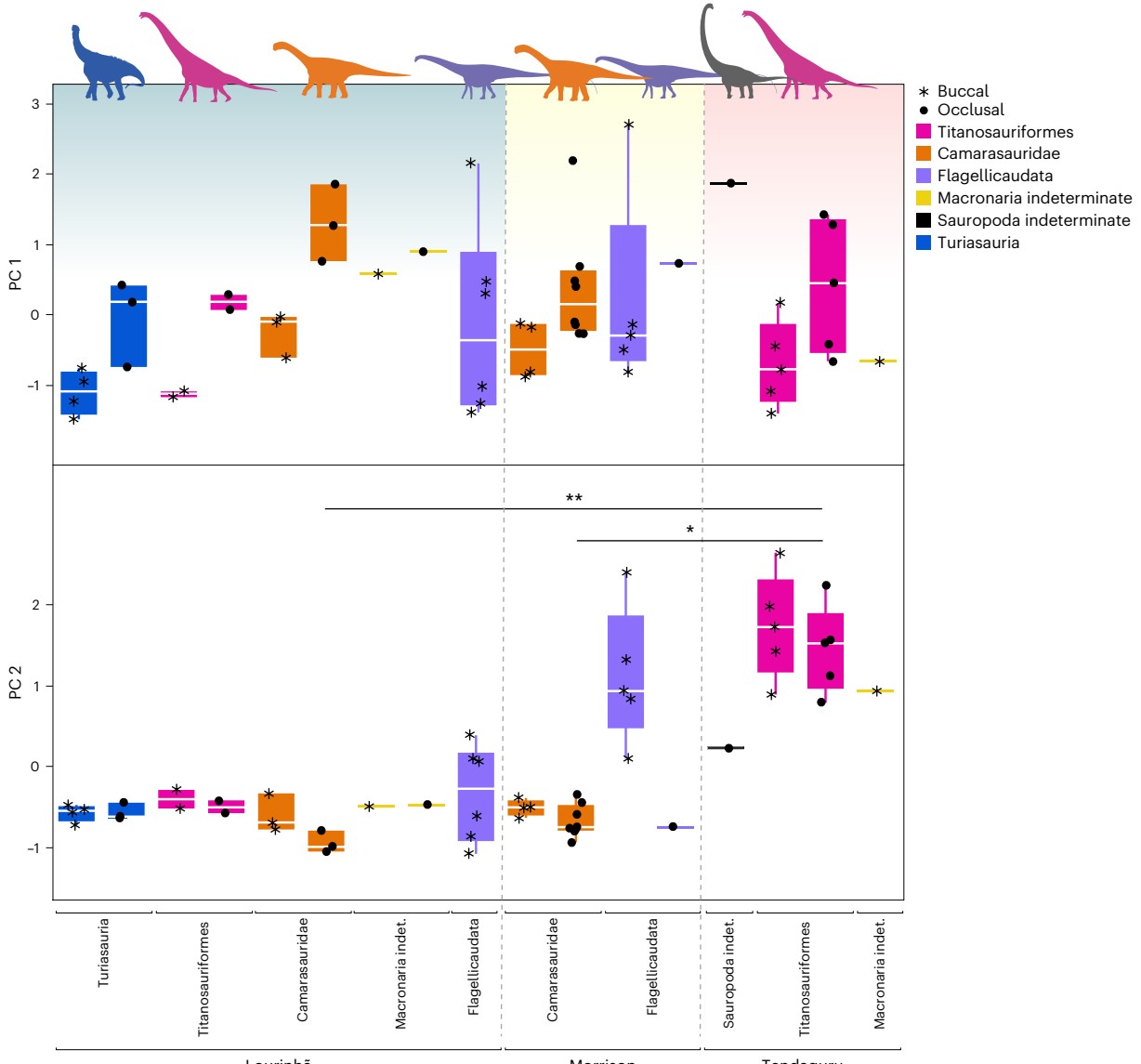

**Fig. 5 | Principal components 1 and 2 for buccal and occlusal surfaces, compared between the three sauropod faunas.** PC 1 represents surface roughness in terms of height and volume, whereas PC 2 reflects structure of the surface in terms of wear pattern complexity and density of wear marks. Boxplots are depicted with a thick horizontal bar that represents the median; the box encloses the first (25%) and third (75%) quartiles; the whiskers extend to the full interquartile range. Significance according to Dunn's pairwise comparison with Bonferroni adjustment (Supplementary Table 4). Number of surfaces per taxon with number of specimens (in case buccal and occlusal surface of the same specimen were included) given in parenthesis: Turiasauria, $n = 5$ (7);

Titanosauriformes, $n = 9$ (15); Camarasauridae, $n = 12$ (20); Flagellicaudata, $n = 12$ (13); Macronaria indet., $n = 3$ (4); and Sauropoda indet., $n = 1$ (1). Level of significance: ***$P = 0.001$, **$P = 0.01$, *$P = 0.05$. Credits: Silhouettes are from Phylopic (https://phylopic.org). *Xinjiangtitan shanshanesis* (Sauropoda indet.), created by Jagged Fang Designs under a CC0 1.0 license; *Amanzia greppini* (Turiasauria), created by T. Dixon under a CC BY 4.0 license; *Diplodocus carnegii* (Flagellicaudata), created by S. Hartman under a CC BY 3.0 license; *Giraffatitan brancai* (Titanosauriformes), created by S. Hartman under a CC BY 3.0 license; *Camarasaurus supremus* (Camarasauridae), created by M. Wedel under a CC BY 3.0 license.

or Tendaguru sauropods ingested much higher abrasive loads compared with sauropods from the other two ecosystems, and especially in Portugal, where titanosauriforms showed distinctly lower surface roughness and complexity than in Tanzania (Fig. 6).

Although the flora of the Tendaguru Formation is different from that of the Lourinhã and Morrison Formations, in the absence of horsetails and pinacean conifers, and the low abundance of cycads[19–21], several lines of evidence suggest that this was not the primary cause for the comparatively strong abrasion in Tendaguru sauropod teeth. Intrinsic abrasives in plants are mostly hard parts, such as seeds and silica-rich phytoliths. In the Tendaguru ecosystem, plants with extant relatives that have a high silica content were ginkgoes, cycads and possibly some ferns[21,66]. These were rather minor components of the

flora, however[21], with ginkgoes mostly growing in conifer-dominated forests[20]; conifers are relatively low in phytolith content[66]. We would therefore not expect overall stronger abrasion in an ecosystem with a lower relative abundance of highly abrasive plants, as is the case in Tendaguru. Moreover, given that titanosauriforms (the only clade we can compare directly between Tendaguru and another biome, in this case Lourinhã) had relatively broad snouts and reached enormous body masses, they were probably rather unselective in their diet, and it is therefore unlikely that they focused on these minor and silica-rich components of the flora. The presence of brachiosaurids and other early-branching titanosauriforms with similar morphologies in all three biomes analysed herein furthermore suggests that food preferences of these sauropods were comparable in the different biomes.

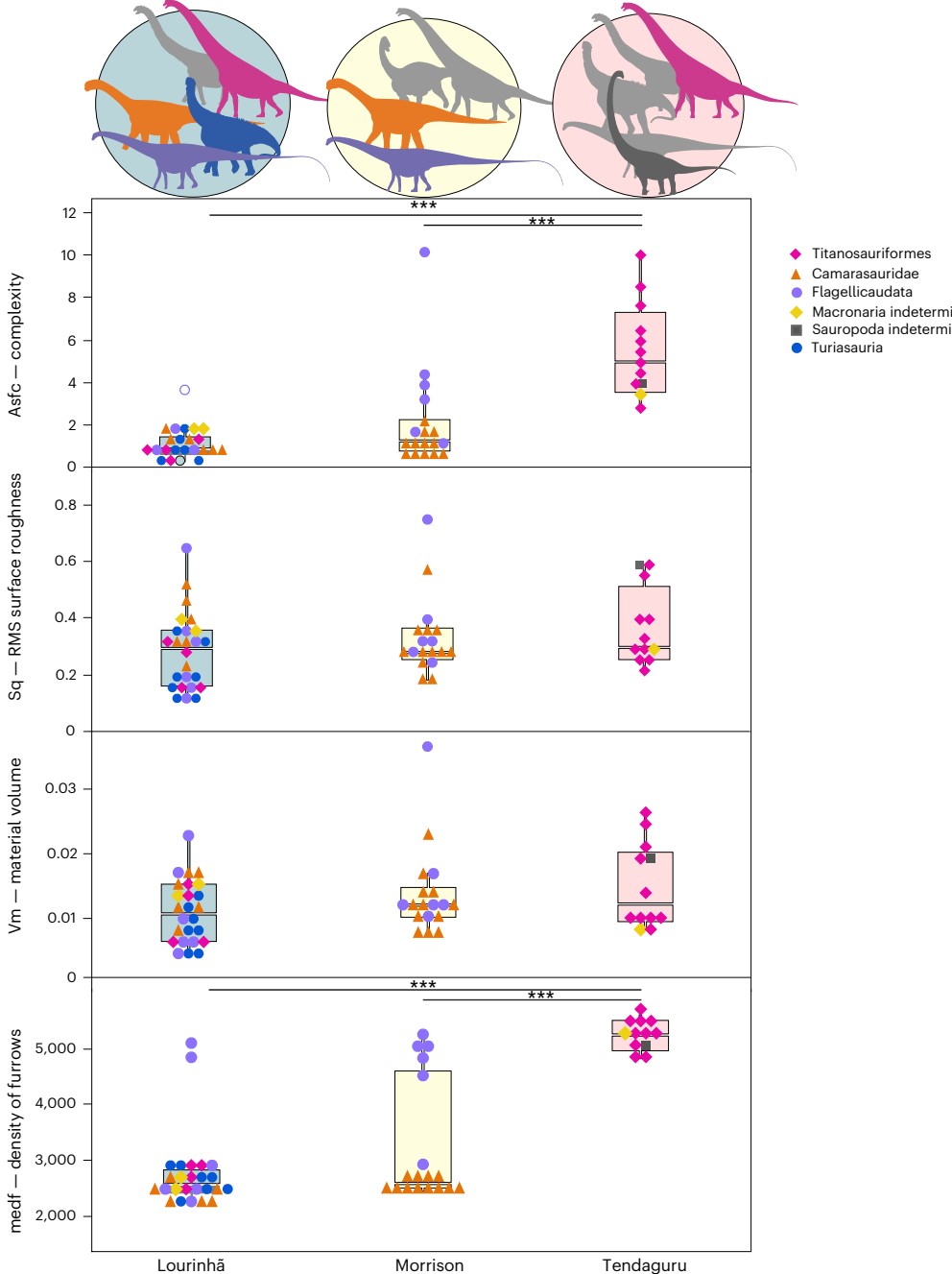

**Fig. 6 | Selected DMT parameter representing complexity, height, volume and density for buccal and occlusal surfaces pooled and for each fauna.** Boxplots are depicted with a thick horizontal bar that represents the median; the box encloses the first (25%) and third (75%) quartiles; the whiskers extend to the full interquartile range. Significance according to Dunn's pairwise comparison with Bonferroni adjustment for multiple comparisons (Supplementary Table 5). Number of surfaces per taxon with number of specimens (in case buccal and occlusal surface of the same specimen were included) given in parenthesis: Turiasauria, $n = 5$ (7); Titanosauriformes, $n = 9$ (15); Camarasauridae, $n = 12$ (20); Flagellicaudata, $n = 12$ (13); Macronaria indet. $n = 3$ (4); and Sauropoda indet., $n = 1$ (1). Level of significance: ***$P = 0.001$, **$P = 0.01$, *$P = 0.05$. Credits:

Silhouettes are from Phylopic (https://phylopic.org). *Xinjiangtitan shanshanesis* (Mamenchisauridae), created by Jagged Fang Designs under a CC0 1.0 license; *Haplocanthosaurus priscus*, created by T. M. Keesey under a CC0 1.0 license; *Amanzia greppini* (Turiasauria), created by T. Dixon under a CC BY 4.0 license; *Euhelopus zdanskyi* (Somphospondyli), created by DiBgd and modified by T. M. Keesey under a CC BY-SA 3.0 license; *Diplodocus carnegii* (Flagellicaudata), created by S. Hartman under a CC BY 3.0 license; *Giraffatitan brancai* (Brachiosauridae, Titanosauriformes), created by S. Hartman under a CC BY 3.0 license; *Camarasaurus supremus* (Camarasauridae), created by M. Wedel under a CC BY 3.0 license.

The distinct abrasion patterns in sauropod teeth from Tendaguru are thus unlikely to be produced by plant-intrinsic abrasives. In fact, such intrinsic abrasives have already been shown to have a lesser impact on dental wear than external mineral abrasives.

Data from mammals[67–71] suggest that external mineral abrasives incorporated with the diet have a more pronounced effect on observed

DMT signatures than plant type and phytolith content. Feeding experiments have shown that depth and complexity of wear features in particular, as well as density of furrows, were more strongly impacted by diets with high external abrasive load than by diets with high phytolith content. The observed distinct differences in complexity (Asfc) and density of furrows (medf) in Tanzanian brachiosaurids hence are

probably stemming from a certain environmental grit load, rather than just phytoliths as internal abrasives of ingesta.

Considering that the Tendaguru ecosystem in Tanzania was a riverine environment featuring extended floodplain areas, a potential source of grit would have been forage gathered close to the ground[35,41]. Low-level forage always comes with substantial amounts of sand and grit attached, if either pulled out of the ground or eaten shortly after rainfall on sandy ground. In extant herbivorous ungulates, soil intake can be up to 33% of dry matter intake in sheep[72,73] or 0.5–1.5 kg in cattle[74]. However, our measurements from Tendaguru are mostly from brachiosaurids, which have obvious adaptations for high-level foraging[27,29,34]. Moreover, the Tendaguru fauna includes relatively abundant remains from the flagellicaudatans *Dicraeosaurus* and *Tornieria*, which have been consistently reconstructed as a low-to-mid-level feeders[29,35,52], but no sampled flagellicaudatan tooth from Tendaguru survived our quality checks[49,50]. The higher abrasive load in Tendaguru brachiosaurids compared with all sauropod teeth except for two teeth referred to Flagellicaudata from the Lourinhã Formation in Portugal (Fig. 6) therefore rather stems from the geographical location of the two ecosystems.

Palaeogeographic reconstructions suggest that Tendaguru would have been near an enormous desert belt, which stretched approximately between palaeolatitudes 16° and 32° S (ref. 75). The presence of this desert probably resulted in wind-blown particles frequently covering the vegetation, which in turn caused more pronounced abrasive wear. In fact, sand-sized external quartz abrasives are known to cause large complexity and roughness values in mammalian herbivores[69,71] that exceed parameter values attributed to high phytolith load. The recovered DMT patterns in Tendaguru sauropods fit this observation (Figs. 2 and 6). Consequently, we think that the hypothesis of the nearby desert as the main contributor of grit to the abrasive load of the Tendaguru environment explains our observed patterns best.

## Conclusion

DMTs can be impacted by a number of factors related to intrinsic features of the food the individual had ingested, the general external abrasive load in the environment and the behaviour of the feeding individual. Teasing apart these factors in fossil ecosystems is not straightforward. In fact, our recovered within-taxon differences across different ecosystems and climate regimes indicate that inferences based on a limited sampling must be treated with some caution. Nevertheless, our study also shows that with careful investigation and correlation of microwear data with climate reconstructions, palaeobotanical evidence and morphological feeding adaptations, some of these factors can be identified. In fact, our data suggest ecological competition between broad-crowned taxa based on distinct patterns in dental microwear. Additionally, low variation in wear features supports earlier reports of migratory behaviour for camarasaurids in the Morrison and Lourinhã Formations, whereas a non-migratory behaviour can be suggested for flagellicaudatans on the basis of highly variable microwear patterns. Moreover, we could detect distinct microwear texture differences in titanosauriforms from Lourinhã and Tendaguru, which we interpreted in terms of a greater load of external abrasives in Tendaguru, probably stemming from a nearby desert. DMTA can thus be a useful additional proxy to palaeogeography and habitat reconstruction.

## Methods

For the 322 scans included in our analysis (Results), we imported a total of 25 of the 34 parameters computed by ref. 50, which are frequently applied on dental wear surfaces for diet inference[47,48,67,69,70,76]. These represent surface height, volume, complexity, density, slope, plateau size and anisotropy (Supplementary Table 6). Larger height and volume parameters (and a greater surface roughness in general) have been found to be indicative of feeding on tougher, more fibrous, or mechanically challenging diets (for example, grasses or hard objects such as

seeds and molluscs[43,48,70,77], but are also affected by greater abrasive loads (dust or grit ingestion)[68,69]. Complexity is a good indicator of hard object feeding[43,77], while anisotropy and density reflect incorporation of small particles (either external or internal abrasives)[70,71,78], but may also be affected by intense, directional oral food processing.

The scan data underwent a visual two-step quality screening by D.E.W. and A.S. to exclude scans affected by sediment particles, glue residues, fractures and postprocessing artefacts. Scans from moulds were also checked for imperfections such as air bubbles. The evaluation was guided by ref. 79 for identifying postmortem dental wear. Each scan was assigned a quality rating: good (1), fair (2) or bad (3); only scans attributed to quality 1 and 2 were considered here. For exemplary surface scans of each clade, see Extended Data Fig. 1.

For each tooth specimen and surface (occlusal and/or buccal), median values per parameter were calculated from up to four (at least one) non-overlapping scans. If several teeth of one individual had been measured, the mean from all teeth was then calculated for the individual. Where more than four measurements were available from different moulds of a single surface, we excluded the scans from the mould that had less scans qualified as quality 1 by ref. 50. For further details on data acquisition and quality assessment, see ref. 50.

Statistical analyses to test for palaeoecological and/or climatic signals in the DMT were conducted in JMP Pro v.17. We used Dunn's test with Bonferroni adjustment for multiple pairwise comparisons. We found pronounced differences between buccal and occlusal surfaces for most clades (Fig. 2 and Supplementary Fig. 1). We therefore analysed buccal and occlusal surfaces both individually and pooled for each clade per fauna (to assess niche partitioning between major taxa). To test for general faunal differences and to assess climate/habitat impact, we solely used the pooled dataset of buccal and occlusal surfaces.

PCA including 20 DMT parameters with a factor loading larger than 0.7 (Asfc, matf, medf, metf, Sa, Sdc, Sdq, Sdr, Sk, Smc, Smr, Sp, Sq, Sv, Sz, Vm, Vmc, Vv, Vvc, Vvv) were computed with varimax factor rotation to facilitate interpretation of principal components (Fig. 3 and Supplementary Table 3). One brachiosaurid specimen from Tendaguru (MB.R. 2190) had to be excluded from the PCA, as it was so aberrant that it affected clear separation of groups in the dataset.

CVA of all 25 parameters was used to maximize separation between clades for detection of niche partitioning, with either clade or clade + fauna as an identifier (Fig. 4 and Supplementary Table 2).

### Reporting summary

Further information on research design is available in the Nature Portfolio Reporting Summary linked to this article.

## Data availability

All measurements used for the analyses herein are included in Supplementary Table 1. They are a subset of measurements exported from a publicly available dataset, which includes the original scan files of tooth surface textures of sauropod teeth more generally (that is, not restricted to the three geological formations we analysed here). The full dataset including all computed measurements based on these scans is available on the UHH Forschungsdatenbank (https://www.fdr.uni-hamburg.de/) at https://doi.org/10.25592/uhhfdm.16992.

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

## Acknowledgements

We thank A. Daasch, K. Engelkes and L. Schwinger (LIB) for technical support during the measurements. We thank the numerous curators

and collection managers (AMNH, CM, CMC, MB.R., ML, NMZ, SMA, USNM and YPM; see ref. 50 for a complete list of people) who have allowed us to sample and/or loan original sauropod teeth. A. Moore (Stony Brook University) provided insights into non-neosauropod tooth morphology. Funding for this work was provided through a Humboldt Research Fellowship for Experienced Researchers (USA 1218977 HFST-E) and a UHH Close the Gap Grant (both to E.T.); a grant through Synthesys+Transnational Access, 4th call (DE-TAF-TA4-009); a doctoral grant from FCT—Fundação para a Ciência e a Tecnología, IP, (2020.05854.BD); and additional support by FCT—Fundação para a Ciência e a Tecnología, IP, through the Research Unit UIDB/04035/2020 (GeoBioTec; https://doi.org/10.54499/UIDB/04035/2020) and the Project PTDC/CTA-PAL/2217/2021 (BioGeoSauria) (all to A.S.).

## Author contributions

D.E.W.: supervision, data analysis, interpretation, writing—original draft, writing—review and editing, visualization. E.T.: project design, supervision, interpretation, writing—original draft, writing—review and editing, visualization, funding. A.S.: data collection, data analysis, writing—original draft, funding. R.W.: data collection, methods. T.M.K.: resources, methods, data analysis, writing—review and editing. D.E.W. and E.T. contributed equally to this work.

## Funding

## Competing interests

The authors declare no competing interests.

## Additional information

**Extended data** is available for this paper at https://doi.org/10.1038/s41559-025-02794-5.

**Correspondence and requests for materials** should be addressed to Daniela E. Winkler or Emanuel Tschopp.

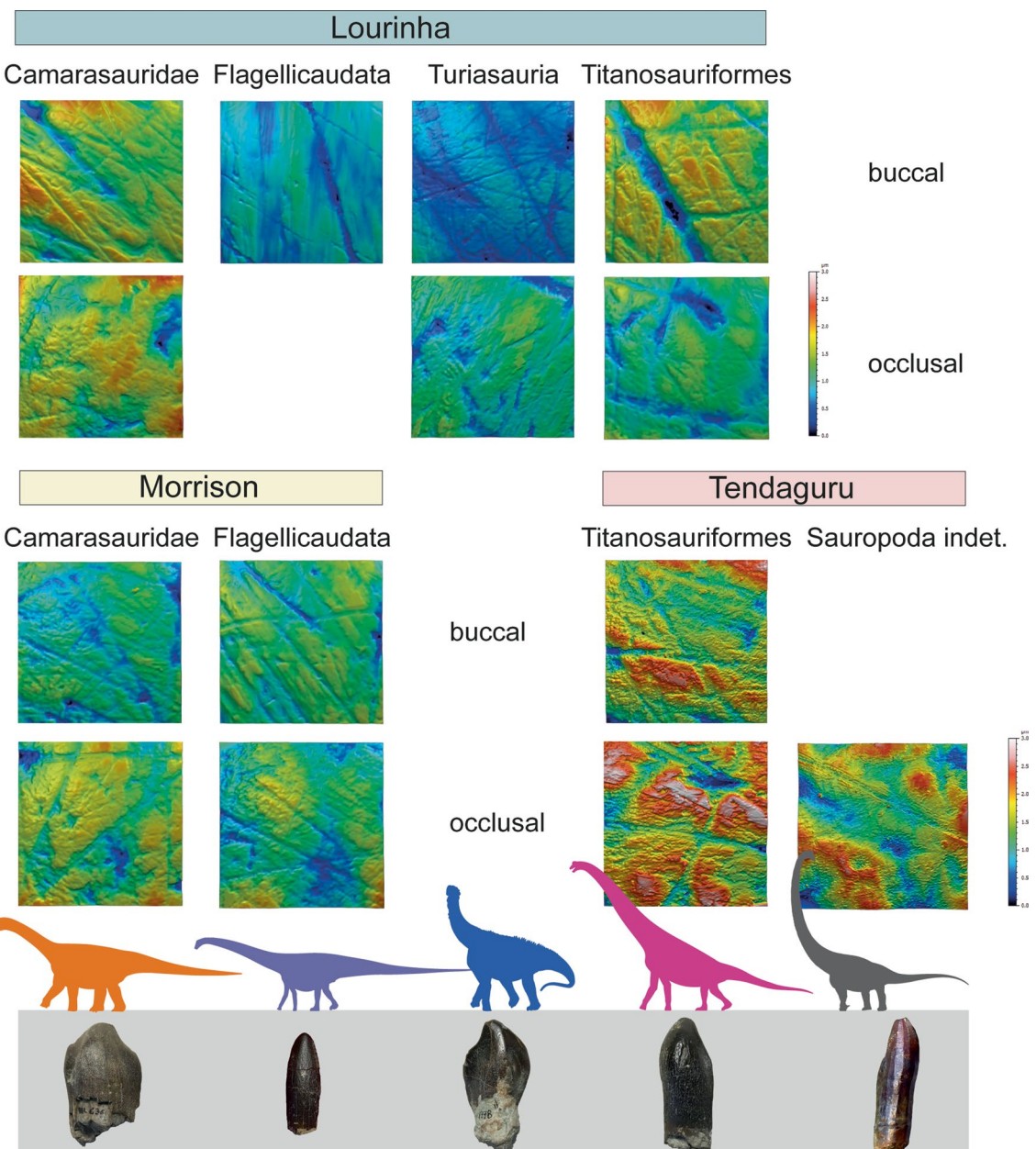

**Extended Data Fig. 1 | Exemplary teeth and 2D images of enamel wear patterns of the analysed clades.** Colour maps represent surface topography: warm colours (red/yellow) indicate higher elevations, while cool colours (green/blue) indicate depressions. All surfaces are to the same scale. Exemplary teeth of the five represented clades are shown in buccal view (Camarasauridae, ML 636; Flagellicaudata, ML 2560; Turiasauria, ML 1178; Titanosauriformes, ML 945, all from Lourinhã, taken from ref. 7; Sauropoda indet., MB.R.2185, from Tendaguru, photo by ET). Credits: Silhouettes are from Phylopic (https://phylopic.org). *Xinjiangtitan shanshanesis* (Sauropoda indet.), created by Jagged Fang Designs under a CC0 1.0 license; *Amanzia greppini* (Turiasauria), created by T. Dixon under a CC BY 4.0 license; *Diplodocus carnegii* (Flagellicaudata), created by S. Hartman under a CC BY 3.0 license; *Giraffatitan brancai* (Titanosauriformes), created by S. Hartman under a CC BY 3.0 license; *Camarasaurus supremus* (Camarasauridae), created by M. Wedel under a CC BY 3.0 license.

# Reporting Summary

## Statistics

For all statistical analyses, confirm that the following items are present in the figure legend, table legend, main text, or Methods section.

| n/a | Confirmed | |
|---|---|---|
| ☐ | ☒ | The exact sample size (*n*) for each experimental group/condition, given as a discrete number and unit of measurement |
| ☐ | ☒ | A statement on whether measurements were taken from distinct samples or whether the same sample was measured repeatedly |
| ☐ | ☒ | The statistical test(s) used AND whether they are one- or two-sided<br>*Only common tests should be described solely by name; describe more complex techniques in the Methods section.* |
| ☒ | ☐ | A description of all covariates tested |
| ☐ | ☒ | A description of any assumptions or corrections, such as tests of normality and adjustment for multiple comparisons |
| ☐ | ☒ | A full description of the statistical parameters including central tendency (e.g. means) or other basic estimates (e.g. regression coefficient) AND variation (e.g. standard deviation) or associated estimates of uncertainty (e.g. confidence intervals) |
| ☒ | ☐ | For null hypothesis testing, the test statistic (e.g. *F*, *t*, *r*) with confidence intervals, effect sizes, degrees of freedom and *P* value noted<br>*Give P values as exact values whenever suitable.* |
| ☒ | ☐ | For Bayesian analysis, information on the choice of priors and Markov chain Monte Carlo settings |
| ☒ | ☐ | For hierarchical and complex designs, identification of the appropriate level for tests and full reporting of outcomes |
| ☒ | ☐ | Estimates of effect sizes (e.g. Cohen's *d*, Pearson's *r*), indicating how they were calculated |

*Our web collection on statistics for biologists contains articles on many of the points above.*

## Software and code

Policy information about availability of computer code

| Data collection | Sensofar optical profiler equipped with software SensoSCAN v.6.7 |
|---|---|
| Data analysis | DMTA was conducted in MountainsMap v.9.1, Besancon, France; statistics were conducted in JMP pro v.17 |

For manuscripts utilizing custom algorithms or software that are central to the research but not yet described in published literature, software must be made available to editors and reviewers. We strongly encourage code deposition in a community repository (e.g. GitHub). See the Nature Portfolio guidelines for submitting code & software for further information.

## Data

Policy information about availability of data

All manuscripts must include a data availability statement. This statement should provide the following information, where applicable:

- Accession codes, unique identifiers, or web links for publicly available datasets
- A description of any restrictions on data availability
- For clinical datasets or third party data, please ensure that the statement adheres to our policy

> All measurements used for the analyses herein are included in the supplementary materials. Original scan files, based on which the measurements were taken are available on the UHH Forschungsdatenbank (https://www.fdr.uni-hamburg.de/) at https://doi.org/10.25592/uhhfdm.16175

# Research involving human participants, their data, or biological material

Policy information about studies with human participants or human data. See also policy information about sex, gender (identity/presentation), and sexual orientation and race, ethnicity and racism.

| Reporting on sex and gender | no human participants in this study |
| --- | --- |
| Reporting on race, ethnicity, or other socially relevant groupings | n/a |
| Population characteristics | n/a |
| Recruitment | n/a |
| Ethics oversight | n/a |

Note that full information on the approval of the study protocol must also be provided in the manuscript.

# Field-specific reporting

Please select the one below that is the best fit for your research. If you are not sure, read the appropriate sections before making your selection.

☐ Life sciences   ☐ Behavioural & social sciences   ☒ Ecological, evolutionary & environmental sciences

For a reference copy of the document with all sections, see nature.com/documents/nr-reporting-summary-flat.pdf

# Ecological, evolutionary & environmental sciences study design

All studies must disclose on these points even when the disclosure is negative.

| Study description | Analysis of mechanical dental wear features for diet inferences (dental microwear texture analysis) of sauropods from three different faunas was conducted to infer niche partitioning, competition, and potential migration patterns. |
| --- | --- |
| Research sample | The sample consists of sauropod teeth housed in different museum collections in the USA, Portugal, and Germany: in total it includes 39 sauropod individuals, 17 of which were recovered from the Lourinha Formation (Portugal), 13 from the Morrison Formation (USA), and nine from the Tendaguru Formation (Tanzania, curated at the Berlin Museum of Natural History) |
| Sampling strategy | When available, we measured wear texture directly on the original teeth, after cleaning the enamel surfaces with acetone and/or ethanol. If not available for loan, tooth specimens were moulded with high resolution dental silicone (Provil novo Light C.D.2 fast set EN ISO 4823, type 3 light, Heraeus Kulzer GmbH, Dormagen, Germany). |
| Data collection | Surface scanning was conducted at the Leibniz Institute for the Analysis of Biodiversity Change (LIB) in Hamburg, Germany. We used the 3D profiling microscope Sensofar S neox (Sensofar-Tech, SL, Terrassa, Barcelona, Spain), with a blue LED (460 nm), vertical resolution of 0.07 µm, spatial resolution of 0.14 µm (in x, y) and numerical aperture of 0.9. The CCD camera resolution was 1232 x 1028 pixels, and the resulting scan size using a 100x objective was 175.44 x 132.10 µm. Each scan was manually cropped to 100 x 100 µm to exclude edge effects and damages present in several scans. |
| Timing and spatial scale | Surface scanning was conducted between June 2022 and November 2023 by AS and RW. |
| Data exclusions | All resulting visual data was put through two manual quality screenings, first done by AS, and then followed by DW. This assessment was done in order to exclude any surface scan and correlating measurements that still remained impacted by foreign structures. These features included sediment particles, along with consolidant and glue residues, and fractures of the enamel surface. Scans from molds and casts were checked for extra features, such as bubbles that formed during the molding and casting process. Furthermore, some scan surfaces included significant portions with plateau-like structures resulting from the digital infilling of holes during the data post-processing step. These scans were also deemed unusable for subsequent statistical analysis, because the large, completely flat areas would have significantly impacted overall texture quantification. As a result of the manual screening, each scan was attributed a value representative of its quality. The three levels created to indicate the scan quality are as follows: quality 1, good; quality 2, fair; and quality 3, bad. A total of 620 scans were deemed to be of good quality, whereas 27 were considered to be of fair quality. We identified 364 scans of bad quality, which should be excluded from future analysis. The majority of good scans resulted from 431 measurements taken from original teeth, followed by 182 taken from molds, and only 7 scans resulting from casts. Scans deemed of fair quality included 8 from original teeth, and 19 resulting from molds, with no scan from a cast falling within this quality level. |
| Reproducibility | Dental microwear texture analysis can be repeated using the same specimens, but re-scanning exactly the same surface area is hardly possible, as teeth or moulds are manually oriented on the microscope plate under the objective. As the measurement process and data analysis are automated, and all scripts for data treatment (filtering) are available, the repeatability is still higher than in 2D microwear studies. |

| Randomization | Groups were created according to taxonomic affinity |
|---|---|
| Blinding | Data analysis was done by DEW on specimens only identified by their inventory number, not with full taxonomic information. |

Did the study involve field work? ☐ Yes ☒ No

# Reporting for specific materials, systems and methods

We require information from authors about some types of materials, experimental systems and methods used in many studies. Here, indicate whether each material, system or method listed is relevant to your study. If you are not sure if a list item applies to your research, read the appropriate section before selecting a response.

## Materials & experimental systems

| n/a | Involved in the study |
|---|---|
| ☒ ☐ | Antibodies |
| ☒ ☐ | Eukaryotic cell lines |
| ☐ ☒ | Palaeontology and archaeology |
| ☒ ☐ | Animals and other organisms |
| ☒ ☐ | Clinical data |
| ☒ ☐ | Dual use research of concern |
| ☒ ☐ | Plants |

## Methods

| n/a | Involved in the study |
|---|---|
| ☒ ☐ | ChIP-seq |
| ☒ ☐ | Flow cytometry |
| ☒ ☐ | MRI-based neuroimaging |

## Palaeontology and Archaeology

| Specimen provenance | All material included in this study is already curated in the following museums:<br>AMNH FARB: American Museum of Natural History; Fossil Amphibian, Reptile, and Bird Collection, New York City, USA<br>CM: Carnegie Museum of Natural History, Pittsburgh, USA<br>CMC: Cincinnati Museum Center, Museum of Natural History & Science, Cincinnati, USA<br>LIB: Leibniz-Institut zur Analyse des Biodiversitätswandels, Bonn and Hamburg, Germany<br>MB.R.: Museum für Naturkunde, Berlin, Germany<br>ML: Museum of Lourinhã, Portugal<br>NMZ: Natural History Museum of the University of Zurich, Switzerland<br>SMA: Sauriermuseum Aathal, Switzerland<br>USNM: Smithsonian National Museum of Natural History, Washington DC, USA<br>YPM: Yale Peabody Museum, New Haven, USA<br>We therefore obtained permisson to include the samples into our analysis from the curators. |
|---|---|
| Specimen deposition | see museums listed above |
| Dating methods | no new dates were calculated |

☐ Tick this box to confirm that the raw and calibrated dates are available in the paper or in Supplementary Information.

| Ethics oversight | no ethical oversight was required, as the study only used museum collection material |
|---|---|

Note that full information on the approval of the study protocol must also be provided in the manuscript.

## Plants

| Seed stocks | no plants involved in this study |
|---|---|
| Novel plant genotypes | n/a |
| Authentication | n/a |

