## [Peer Review File · Nature Ecology & Evolution]

Dental microwear texture analysis reveals behavioural, ecological, and habitat signals in Late Jurassic sauropod dinosaur faunas

Corresponding Author: Dr Emanuel Tschopp

Version 0:

Decision Letter:

21st January 2025

Dear Emanuel,

Your manuscript entitled "Dental microwear texture analysis (DMTA) reveals behavioral, ecological, and habitat signals in Late Jurassic sauropod dinosaur faunas" has now been seen by three reviewers, whose comments are attached. The reviewers have raised a number of concerns which will need to be addressed before we can offer publication in Nature Ecology & Evolution. We will therefore need to see your responses to the criticisms raised and to some editorial concerns, along with a revised manuscript, before we can reach a final decision regarding publication.

As you will see from the reviews, the reviewers found it difficult to access all the information they needed to verify/reproduce the results of the manuscript--please could you pay particular attention to these queries so that they can fully assess those aspects in the next round of revision. If the files are not yet available via OSF, I would suggest using a temporary external repository such as figshare or morphosource (or both) to share the files for the purposes of review.

We therefore invite you to revise your manuscript taking into account all reviewer and editor comments. Please highlight all changes in the manuscript text file.

* If you have not done so already please begin to revise your manuscript so that it conforms to our Article format instructions at <http://www.nature.com/natecolevol/info/final-submission>. Refer also to any guidelines provided in this letter.

* Extended Data Figures - please ensure that any supplementary figures and tables that are crucial to the manuscript's conclusions are converted into Extended Data figures and tables to increase visibility of these data. Extended Data figures and tables are online-only (present in the online PDF and full-text HTML versions of the paper), peer-reviewed display items that provide essential background to the article but are not included in the main article due to space constraints. A maximum of ten Extended Data display items (figures and tables) is permitted.

Link Redacted

Nature Ecology & Evolution is committed to improving transparency in authorship. As part of our efforts in this direction, we are now requesting that all authors identified as 'corresponding author' on published papers create and link their Open Researcher and Contributor Identifier (ORCID) with their account on the Manuscript Tracking System (MTS), prior to acceptance. ORCID helps the scientific community achieve unambiguous attribution of all scholarly contributions. You can create and link your ORCID from the home page of the MTS by clicking on 'Modify my Springer Nature account'. For more information please visit www.springernature.com/orcid.

[redacted]

Reviewer expertise:

Reviewer #1: signed review

Reviewer #2: signed review

Reviewer #3: reptile dental microwear

Reviewers' comments:

Reviewer #1 (Remarks to the Author):

I find this an interesting and important study of the effect of climatic and environmental differences on sauropod tooth wear, a topic that has so far been hardly addressed. The manuscript and the figures and tables are mostly well prepared. Some of the figures would be easier to read with larger font size. In part, I think the results could be clarified in the main text and figure captions, for example in terms of how the results of the statistic analyses are shown (now they are just presented in the supplementary tables, which are hardly referenced in the manuscript text or figure captions). The results themselves are very interesting. Some of the discussion on how to interpret the results seems a bit convoluted and speculative, although it arrives at a plausible conclusion that the more abrasive signal in sauropod tooth wear in Tendaguru was probably caused by windblown dust from the deserts nearby. However, this reasoning could be clarified in my mind. First, I find the speculation about brachiosaurid consumption of gritty plant roots very unlikely, as such a behavior is not usual in most present-day large herbivorous mammals and because the ecomorphology of brachiosaurid sauropods does not suggest such a behavior. Second, it is implied in the text that the microwear texture analysis indicates the presence of abrasion from exogenous grit (windblown dust?) rather than phytoliths, but this is not explained in more detail (i.e., can the method separate the effects of dust and phytoliths in abrasive wear signal?). Furthermore, the role of abrasive particles in the plants themselves is briefly mentioned, with a note that conifers don't produce large amounts of phytoliths. However, it is possible that other kinds of Mesozoic plants (e.g. cycads) could have caused more abrasive tooth wear. Thus, I think it would be interesting to include some comparison of plant communities and availability of plant resources for the sauropods in Morrison, Tendaguru and Lourinha, based on paleobotanical data. This might allow you to better evaluate whether windblown dust is indeed the most likely factor behind the observed differences in sauropod microwear between the localities, or whether plant community differences could have played a role. Paleobotanical data should exist at least for Morrison and Tendaguru (e.g. Gee 2011 and references therein). Find some more comments in the annotated manuscript file attached here. I recommend this manuscript to be published after a revision.

Sincerely,
Juha Saarinen

Reference: Gee, C. T. 2011. Dietary options for the sauropod dinosaurs from an integrated botanical and paleobotanical perspective. In: Biology of the sauropod dinosaurs: Understanding the life of giants. N. Klein, K. Remes, C.T. Gee and P. M. Sander(eds.), Indiana University Press, Bloomington.

Reviewer #2 (Remarks to the Author):

The manuscript "Dental microwear texture analysis (DMTA) reveals behavioral, ecological, and habitat signals in Late

Jurassic sauropod dinosaur faunas" by Winkler, Tschopp et al. presents hypotheses on feeding niches and migration patterns of sauropods from three well-known Late Jurassic areas. They perform very detailed analyses thanks to a very large sample of tooth microwear patterns obtained in a previous work. It is a work with a good bibliographical basis on which the authors support their results to elaborate the discussion and hypotheses mentioned above. The figures, tables and supplementary material complement the work, and help to understand the text, which is already easy and enjoyable to read. I have no further comments to make, just two suggestions that I think could help the understanding of the methodology:

- I understand that Saleiro et al. (2024) describe in great detail the whole DMTA methodology to obtain the dataset used in this work, and that it would be redundant information here, but for readers who see this methodology for the first time I think it would be very useful to summarise (if possible) in one or two sentences what it consists of, also to increase the interest in reading Saleiro et al. (2024).

- I have seen that in both papers the authors mention a tooth tentatively referred to a mamenchisaurid from the Tanzanian sample. It would be very interesting if they could mention a bit more in detail the reasons why the authors think that this tooth could belong to this clade, since until now only a caudal series has been referred to the only mamenchisaurid taxon recognized in Tendaguru.

I have added a couple of other details in the pdf. I found it a very interesting work and hope to see it published soon.

Dr. Verónica Díez Díaz

Reviewer #3 (Remarks to the Author):

The authors present a detailed comparison of tooth wear patterns using DMTA on teeth from several sauropod taxa from three Late Jurassic localities (USA, Portugal, and Tanzania). Their comparisons of various metrics indicative of differential tooth wear patterns and - therefore - some aspect of diet lead the authors to make several conclusions regarding the potential niche partitioning of co-occurring sauropods, and the potential for migration in camarasaurids in particular. In general I found the discussion to be of interest, but I did find it hard to follow the rationale of this study throughout, and I think this has more to do with data visualization and clarity of the methods. Moreover, I was concerned with the limited sampling of certain taxa in some localities where broad, sweeping conclusions later in the paper made it seem like a statistically well-supported conclusion. Overall there is potential here, but I found two broad areas of concern that would have greatly improved the manuscript and made it easier to follow. A lot of the text makes assumptions that the reader is familiar with DMTA and the well-established signals for seasonality and diet that can be inferred from these results for extinct herbivorous reptiles. So for this reason, I have many questions/comments that I hope will improve the clarity of the manuscript. They fall into two broad categories below, and I raise many points in the appended PDF.

(1) There is a lack of clarity in the materials and methods, and no representation of any of the raw data:

I found myself having to read and re-read the materials and methods and results several times to follow the specimen and taxon sampling for this study. Part of the reason is that the supplementary data table divides specimens into # of teeth, # of scans, and other iterations that make it a bit hard to visualise how many individual specimens vs teeth are being discussed and compared. Some of this is clarified in the figures, but I would just like to know in the main text how many isolated teeth vs individual specimens (which could have multiple teeth to sample) are compared here. This is hinted at in the supplementary table, but it is important because each tooth has its own ontogenetic trajectory relative to each specimen. An animal would have multiple lightly worn, heavily worn, and unworn teeth in its mouth at any given time, which would (I assume) affect all of the metrics the authors were concerned with. So were teeth selectively sampled if they showed heavier tooth wear, or were teeth scanned/molded non-discriminately along the jaws? Or were these mostly isolated teeth where the authors had no control over how worn a tooth was? These are sampling factors that become important when talking about the many inferences in the discussion.

And were you always comparing patterns of enamel wear across specimens or does this also extend into the dentine? Surely many sauropods wore through the enamel and into the dentine during the life of each tooth. Dentine is much softer and more elastic and is therefore prone to different wear mechanics. How was this difference accounted for in the analyses? I would really have liked to see representative images of the different tooth wear patterns of interest in the main text figures. In fact, any images of the actual raw data. These are nowhere to be found in the main text or supplementary information, correct? The teeth and their wear facets are the raw data, so several of these and their different wear patterns should be illustrated in the main text.

The other concern here is the repeated reference to Saleiro et al. [47], which as I now understand it is a preprint that contains information about the sampling protocol and images of specimens. This is the sort of thing I was looking for in the materials and methods, supplementary information, or in Figure 1 or 2 to help me understand what the authors did. Instead, the data collection protocol, the various parameters of interest, and several other facets of the study design rely on a preprint rather than standing on its own in a self-contained manuscript. I am against this. I think the methods from the preprint should be repeated or combined somehow into this manuscript and be subject to peer review. For example, I cannot judge what "complexity" or "density of furrows" refers to as a non-specialist on DMTA. All of this needs more clarity for a general scientific audience.

The same applies to the 3-D scans. These are supposedly being made available, but as of Jan. 2025, I cannot access them

(I followed the various links). Please ensure all of the data used for this are accessible for review. As it stands, I cannot tell whether I agree with how a particular feature was measured or not, because it is impossible to tell.

(2) Comparisons across localities are confounded with uneven taxon sampling:

There is partial overlap of some elements of the sauropod fauna across the three localities (USA, Portugal, Tanzania), making it difficult to draw sweeping conclusions, but even more importantly, sample size for given taxa of interest are wildly different in some cases. In particular, Brachiosaurids are fairly(ish) well-represented from Portugal, but there are only 2 measurements from Tanzania. At times the text is cautious about drawing too many conclusions from limited data, but that is not the case in the abstract or conclusions. There it is an oversell at times and I highlight this in the attached PDF.

First- and second-order inferences regarding niche partitioning, competitive exclusion, migration, and foraging behaviours throughout the discussion had me going back to the original dataset repeatedly. I am certain these measurements are difficult to take and collections access is always a challenge, but it seems as though the partial overlap of sauropod assemblages across these three localities, coupled with limited sample sizes for some groups, make the discussion weaker.

For example, the comparisons between brachiosaurids sound at first glance to be important, but the underlying numbers of samples are concerning. In the end, this is a comparison between 9 samples from Tanzania, and 2 from Portugal? I understand that the effect size of your given metrics of choice for comparison may be high, but this is still a troublingly small sample size for comparisons, particularly from Portugal. And what about the supposed migratory or feeding habits of Portuguese camarasaurids? This a lot of space dedicated to the potential migratory signal of camarasaurids based on tooth wear measurements from Morrison Formation samples, but what about Portugal? Is this a consistent signal across the group? Are these similar types of camarasaurids that are being compared?

*****END*****

Version 1:

Decision Letter:

9th May 2025

Dear Emanuel,

Thank you for submitting your revised manuscript "Dental microwear texture analysis (DMTA) reveals behavioural, ecological, and habitat signals in Late Jurassic sauropod dinosaur faunas" (NATECOLEVOL-24113333A). It has now been seen again by the original reviewers and their comments are below. The reviewers find that the paper has improved in revision, and therefore we'll be happy in principle to publish it in Nature Ecology & Evolution, pending minor revisions to satisfy the reviewers' final requests and to comply with our editorial and formatting guidelines. Please note however that this will involve making the data needed to support the paper's conclusions available to comply with our data policy (<https://www.nature.com/nature-portfolio/editorial-policies/reporting-standards>). It would be helpful if you could let me know what the latest status of the underlying data preprint is--as reviewer 3 mentions, "the lack of actual images of the fossils or maps themselves is an issue that needs to be addressed in this manuscript". We are happy for authors to make use of separate DOIs for datasets, however these need to be accessible on publication. Failing that, you will need to make the appropriate sub-section of that dataset necessary to support this paper's findings available separately.

If you have not done so already, please ensure that you also email us completed copies of the Reporting summary and Editorial policy checklists:

Reporting summary: <https://www.nature.com/documents/nr-reporting-summary.pdf>

Editorial policy checklist: <https://www.nature.com/documents/nr-editorial-policy-checklist.pdf>

[redacted]

Reviewer #1 (Remarks to the Author):

As far as I can tell, the authors have satisfyingly answered the reviewers' comments and suggestions and revised the manuscript accordingly. I thus recommend the manuscript to be accepted for publication.

Reviewer #2 (Remarks to the Author):

The authors have done a very good job in addressing all the comments suggested by the reviewers and including new information in the text, while correctly discussing those changes that were not implemented. I believe that the manuscript needs no further revisions, and that in conjunction with the work of Saleiro et al. (under review), it will be two very important studies in the study of sauropod wear patterns and feeding habits.

I have only two last thoughts:

- p. 2, line 57: 'The only notable difference between the faunas of the three formations we analyse here was the absence of the conifer clade (...)'. Wouldn't the term to use here be 'floras' instead of 'faunas'?

- In the Extended Data Figure caption: I think it would be good to include an explanation of the colour patterns, e.g.: 'Colour maps represent surface topography: warm colours (red/yellow) indicate higher elevations, while cool colours (green/blue) indicate depressions'.

Reviewer #3 (Remarks to the Author):

The authors have weighed most of my comments in their revised version of the manuscript. I have a few minor corrections and comments in the attached PDF for them to consider. These are mostly quick issues to either rebut or include in a modified version of the manuscript.

I understand the complexities behind dual publications of the same dataset, but the lack of actual images of the fossils or maps themselves is an issue that needs to be addressed in this manuscript. The clearer the authors can be about methodology and raw data representation, the better. I think the new surface texture figure is a good option. I had a minor suggestion of including representative images of these in one of the main text figures, if there is space.

I hope you find the comments helpful.

Kind regards,

Aaron LeBlanc
